# A systematic review of the prevalence of lifetime experience with 'conversion' practices among sexual and gender minority populations

**Travis Salway**[1,2,3]*, **David J. Kinitz**[4], **Hannah Kia**[5], **Florence Ashley**[6], **Dean Giustini**[7], **Amrit Tiwana**[1], **Reilla Archibald**[1], **Amirali Mallakzadeh**[1], **Elisabeth Dromer**[8], **Olivier Ferlatte**[9,10], **Trevor Goodyear**[2,11], **Alex Abramovich**[4,12]

1 Faculty of Health Sciences, Simon Fraser University, Burnaby, British Columbia, Canada, 2 Centre for Gender and Sexual Health Equity, Vancouver, British Columbia, Canada, 3 British Columbia Centre for Disease Control, Vancouver, British Columbia, Canada, 4 Dalla Lana School of Public Health, University of Toronto, Toronto, Ontario, Canada, 5 School of Social Work, University of British Columbia, Vancouver, British Columbia, Canada, 6 Faculty of Law & Joint Centre for Bioethics, University of Toronto, Toronto, Ontario, Canada, 7 Biomedical Branch Library, University of British Columbia, Vancouver, British Columbia, Canada, 8 School of Psychology, University of Ottawa, Ottawa, Ontario, Canada, 9 École de santé publique, Université de Montréal, Montréal, Quebec, Canada, 10 Centre de Recherche en Santé Publique, Université de Montréal et CIUSSS du Centre-Sud-de-l'Île-de-Montréal, Montréal, Quebec, Canada, 11 School of Nursing, University of British Columbia, Vancouver, British Columbia, Canada, 12 Institute for Mental Health Policy Research, Centre for Addiction and Mental Health, Toronto, Ontario, Canada

* travis_salway@sfu.ca

**Data Availability Statement:** The data used in this systematic review are available in a Supplementary File ("Dataset").

## Abstract

### Rationale

Conversion practices (CPs) refer to organized attempts to deter people from adopting or expressing non-heterosexual identities or gender identities that differ from their gender/sex assigned at birth. Numerous jurisdictions have contemplated or enacted legislative CP bans in recent years. Syntheses of CP prevalence are needed to inform further public health policy and action.

### Objectives

To conduct a systematic review describing CP prevalence estimates internationally and exploring heterogeneity across country and socially relevant subgroups.

### Methods

We performed literature searches in eight databases (Medline, Embase, PsycInfo, Social Work Abstracts, CINAHL, Web of Science, LGBTQ+ Source, and Proquest Dissertations) and included studies from all jurisdictions, globally, conducted after 2000 with a sampling frame of sexual and gender minority (SGM) people, as well as studies of practitioners seeing SGM patients. We used the Hoy et al. risk of bias tool for prevalence studies and summarized distribution of estimates using median and range.

**Funding:** TS, Canadian Institutes of Health Research (PCS - 168193) https://cihr-irsc.gc.ca/ The funders had no role in study design, data collection and analysis, decision to publish, or preparation of the manuscript.

**Competing interests:** The authors have declared that no competing interests exist.

## Results

We identified fourteen articles that reported prevalence estimates among SGM populations, and two articles that reported prevalence estimates from studies of mental health practitioners. Prevalence estimates among SGM samples ranged 2%-34% (median: 8.5). Prevalence estimates were greater in studies conducted in the US (median: 13%), compared to Canada (median: 7%), and greater among transgender (median: 12%), compared to cisgender (median: 4%) subsamples. Prevalence estimates were greatest among people assigned male at birth, whether transgender (median: 10%) or cisgender (median: 8%), as compared to people assigned female at birth (medians: 5% among transgender participants, 3% among cisgender participants). Further differences were observed by race (medians: 8% among Indigenous and other racial minorities, 5% among white groups) but not by sexual orientation.

## Conclusions

CPs remain prevalent, despite denouncements from professional bodies. Social inequities in CP prevalence signal the need for targeted efforts to protect transgender, Indigenous and racial minority, and assigned-male-at-birth subgroups.

## Introduction

Conversion practices (CPs) refer to organized attempts to deter participants from adopting or expressing non-heterosexual identities or gender identities that differ from their gender/sex assigned at birth [1–3]. CPs occur within a broader societal context of cissexism and heterosexism—belief systems that position cisgender and heterosexual identities and orientations as preferred and desirable [1]. Cissexism and heterosexism lead to stigma, discrimination, violence, and a lack of protections and access to social determinants of good health for lesbian, gay, bisexual, transgender (trans), and queer (herein, sexual and gender minority [SGM]) people [1]. This, in turn, produces numerous health disparities among SGM people, including unequal access to healthcare—a resource to which access in many settings is already limited by cost [4].

CPs go by many different names, including 'conversion therapy', sexual orientation or gender identity or expression change efforts, and 'reorientation therapy', among others [2, 3]. The variability in nomenclature for CPs makes it challenging to identify and research these practices. Notwithstanding methodological challenges in categorizing and defining CPs, recent international reports demonstrate that these practices continue globally, with heterogeneity within and across jurisdictions, and continue to cause psychosocial harm among SGM people [5, 6]. Demonstrated psychosocial harms associated with exposure to CPs in SGM populations include severe psychological distress, depression, substance abuse, and attempted suicide [7].

Synthesizing CP prevalence estimates is critical to several public health-related efforts. First, quantifying the prevalence of people who have experienced CPs can support advocacy and motivate health professionals and policymakers to implement CP prevention strategies (e.g., legislative bans) and health promotion initiatives for CP survivors. For instance, results from a 2019 Canadian survey demonstrating that as many as 50,000 (10% of) sexual minority men had experienced CP helped to motivate federal politicians to enact legislation to ban CPs (law enacted January 2022) [8, 9]. Second, prevalence estimates are needed to monitor and evaluate

the effectiveness of CP bans and related policies. If bans effectively deter CPs, there should be temporal and jurisdiction-specific reductions in prevalence estimates—a hypothesis that can be tested using quasi-experimental research methods. Third, within-population research of CP prevalence can help identify subgroups most at risk of CP and geographies with high frequency of CP. These subgroup estimates, in turn, can inform tailored, equitable CP prevention and recovery efforts.

A 2022 systematic review focused on youth synthesized prevalence estimates from the United States (US), finding that 12% of SGM people have experienced CPs, in turn contributing to total economic burden of CP-associated harms (e.g., substance use, suicide attempts) of $9.23 billion (USD) [7]. This review excluded estimates from countries other than the US and did not explore heterogeneity in prevalence estimates across studies. To date, no international systematic review has been published describing CP prevalence estimates across geographies, nor has any systematic review explored variability of CP prevalence across socially meaningful subgroups (e.g., gender identity, sexual orientations, race/ethnicity). The number of CP research articles published has increased substantially in the past decade. For example, the frequency of articles indexed in PubMed with either of two common CP-related phrases ("conversion therapy" OR "reparative therapy") increased from 1–4 citations per year before 2010 to 6–79 relevant citations per year between 2010 and 2021. The growth of research on CP may be attributable to a climate of increased human rights protections for SGM populations, particularly in some OECD (Organization for Economic Co-operation and Development) jurisdictions, and in turn, interest in examining the adverse impacts of exposure to non-affirming psychosocial interventions among SGM people [10].

Given this dramatic increase in publications of CP empirical studies and the public health evidence needs cited above, we conducted a systematic review of contemporary (2000 and onward) publications and extracted CP prevalence data to describe the range and median of CP prevalence estimates internationally and explore heterogeneity in estimates by comparing ranges and medians across the country and socially relevant subgroups (i.e., by gender identity, gender modality—i.e., cisgender [cis] or trans, sexual orientation, and race).

## Methods

### Positionality, protocol, and research questions

We came to this study as an interdisciplinary group of health researchers invested in advancing equity- and evidence-informed public policy with SGM people, including in the context of CPs. Our authorship team includes SGM people and allies with expertise in population approaches to SGM health research [1]. For reporting purposes, the present study followed PRISMA 2020 guidelines for systematic reviews and especially the elaboration documents written by the authors [11]. The protocol has been peer reviewed (including independent peer review of search strategy using Peer Review of Electronic Search Strategies), registered with PROSPERO (CRD42020196393, September 18, 2020), and published [12]. Our review includes two distinct but related research questions. First, what is the scope (prevalence) of CP among SGM populations, globally? Second, what is the nature of CP, globally? Given the difference in nature of data (quantitative prevalence estimates for first research question; a mixture of qualitative and quantitative research for the second), the two subsets of articles are summarized separately, and the present article addresses the first research question only.

### Literature search and selection

We performed comprehensive literature searches on January 4, 2022, using the following bibliographic databases: Medline (OVID), Embase (OVID), PsycInfo, Social Work

Abstracts via EBSCO, CINAHL, Web of Science Core Collection, LGBTQ+ Source, and Proquest Dissertations & Theses Global and Sociology Collection. The Medline (OVID) and Embase (OVID) searches were first developed by a biomedical librarian (DG) in conjunction with the research team and adapted to each bibliographic database based on their specific requirements. We developed an exhaustive list of keyword search terms to identify CP across settings and contexts, associated with which is significant variability in terminologies used to refer to CP [12], including variants of "conversion," "repair," and sexual orientation or gender identity "change efforts," among others. This set of CP-related citations was intersected with sets of free text and index terms (e.g., MeSH in Medline) to express the diverse sexual and gender identities of the SGM population, and by adapting a previously validated search strategy population [13]. A sample search strategy is provided in S1 File.

To increase our overall search sensitivity, we supplemented the database searches with hand searches for additional articles using the reference lists of studies retrieved from database searches and from the aforementioned systematic review of the economic burden of CPs [7]. We also conducted targeted grey literature searches of consensus statements issued by health professional bodies [14–16] as well as key websites indexing SGM research (e.g., Human Rights Campaign's resource page regarding CP [12]. Where grey literature reports were subsequently published in peer-reviewed scientific journals (e.g., Trevor Project's National Survey of LGBTQ Mental Health; US Transgender Survey), we opted to include the latter, peer-reviewed sources [17, 18]. Covidence was used to remove duplicates and manage citations.

We applied the following inclusion/exclusion criteria to select articles in two stages (title/abstract screening followed by full-text screening):

- Language: We included articles published in English, Spanish, or French (languages understood by our research team).

- Participants: We included studies with a sampling frame of SGM people, as well as studies of practitioners (e.g., counsellors, psychiatrists) seeing SGM patients/clients/service users.

- Dates: We excluded studies with data collected prior to 2000, given that literature regarding CP prevalence was sparse before this date and the increased opportunities for SGM people to have historically stigmatized identities supported, socially and structurally, post-2000 [19].

- Geography: Given the ubiquity of CP globally [5] and the scarcity of prevalence data, we did not impose any restrictions with regard to geography.

- Study designs: We included empirical studies related to our research question. For the present report, we only included quantitative studies, typically cross-sectional or cohort studies of SGM populations.

We excluded studies that included respondents who were identified as both cis and heterosexual since they are rarely offered CPs [20]. For the present investigation, we additionally excluded samples that were defined in a way that would plausibly bias an estimate of CP prevalence among an entire SGM population. This included studies with a sampling frame focused on people who were likely seeking out CP—for example, samples defined as people seeking "resources for same-sex attracted [members of a faith]" [21, 22]. We also excluded studies that used a measure of CP that included having been recommended to receive CP without separately estimating the number of respondents who actually underwent CP [23]. Two authors (TS, DK, and/or ED) reviewed each title/abstract and full text, and a third author resolved discrepancies that could not be resolved by consensus.

## Data extraction

For each study, we extracted a numerator (number exposed to CP) and denominator (total number sampled), including by subgroups as applicable. Where numerators were not reported, we calculated the numerator by multiplying the proportion exposed to CP by the denominator. We additionally recorded, for each study, the country from which data were collected, which enabled us to examine differences in frequency of CP by jurisdiction. Subgroups were classified according to: gender modality (cis or trans, inclusive of non-binary), gender/sex assigned at birth, sexual orientation identity, and race. Subgroups were used where reported across >1 study. Given that CP targeting trans people aims to make them conform with gender/sex assigned at birth [24], gender identity was coded into four subgroups based on gender/sex assignment at birth and alignment with current gender identity. Gender identity subgroups included cis assigned male at birth (AMAB); cis assigned female at birth (AFAB); trans and/or nonbinary AMAB; and trans and/or nonbinary AFAB. Sexual orientation was coded as asexual, gay/lesbian, and plurisexual—defined as experiencing sexual attraction to people of multiple genders, and inclusive of bisexual, queer, and pansexual people [25]. (Only one study reported the prevalence among heterosexual trans people.) Finally, for studies from Canada, the United Kingdom (UK), and the US, race was coded as Indigenous to North America (Canada and US only); non-Indigenous racialized (inclusive of Black, Latinx/Hispanic, Asian, Multiracial, etc. subgroups); and white. CP prevalence estimates were not divided by race in studies from other countries. We additionally examined age-related patterns within studies, though did not extract age group-specific prevalence estimates because age categorization differed across studies. Some studies reported prevalence estimates by markers of socioeconomic status (e.g., occupation, education, income); however, there was insufficient cross-study consistency in categorization of these variables to synthesize estimates by socioeconomic status.

## Risk of bias assessment

We assessed risk of bias among the studies included in the review in order to systematically describe methodological limitations to the available literature. We used the Hoy et al. risk of bias tool for prevalence studies, given its flexibility across research contexts [26]. The tool was adapted to remove three irrelevant items, as explained in our systematic review protocol [12]. Briefly, items 2–3 were removed because there is no known sampling frame for SGM, making it impossible to assess sample representativeness in relation to target population or randomly select from an SGM frame [27, 28]; item 7 was removed because there are no measures of CP that have been tested for psychometric properties. The remaining seven items were assessed as high or low risk [26]. For item 6 ("acceptable case definition"), we rated a study as having high risk of bias if no definition of CP was provided to study participants. For item 9 ("appropriate prevalence period"), we rated a study as having high risk of bias if any recall period other than 'lifetime' was used.

## Analysis

We anticipated substantial heterogeneity, or variability, in CP prevalence estimates for several reasons. First, there is wide variability in how the included studies defined or measured CP [17, 29]. Second, exposure to CP varies across people and groups affected by intersecting systems of power and oppression, including those related to gender, sexual orientation, race, and class; therefore, we would expect individual CP prevalence estimates to be influenced by the composition of the sample itself [8]. Third, given the reliance of most SGM research on non-probabilistic (i.e., non-random) samples drawn from SGM community venues [30], and the

lack of a defined sampling frame (e.g., census or other enumerated list) of SGM populations [28], samples will vary in the degree to which they capture people with histories of CP, especially people who may not currently endorse a lesbian, gay, bisexual, trans, or queer identity [31].

Given the above, we synthesized prevalence estimates by calculating the median, interquartile range, and range, as recommended by the *Cochrane Handbook* [32]. We further summarized the range and distribution of prevalence estimates stratified by region/country and by the most reported subgroups, including those defined by gender modality, gender modality combined with gender/sex assignment at birth, sexual orientation, and race. Heterogeneity across included studies was quantified using the $I^2$ statistic and Cochran $Q$ test ($p<0.05$ considered statistically significant) [33]. All analyses were completed in RStudio version 1.4.1106. Forest plots and heterogeneity statistics were generated using the metafor package [33].

## Results

### Summary of studies included

A total of 776 unique records were identified across the nine databases. An additional nine records were identified through authors' networks and reference screening. After applying exclusion criteria through title/abstract and full-text screening, 51 articles were identified as relevant and included in the broader systematic review (Fig 1). Sixteen of these articles offered estimates of the prevalence of CP and are included in the present report (synthesis of studies meeting objectives for our second research question is provided in a separate, forthcoming article). Fourteen of the 16 articles report prevalence estimates from 14 distinct empirical studies of SGM populations, while two report prevalence estimates from empirical studies of mental health practitioners.

All 14 of the SGM studies were conducted after 2010. Six were conducted in the US [3, 17, 18, 34–36], four in Canada [8, 29, 37, 38], and one each in: Australia [39], Colombia [40], South Korea [41], and the UK [42]. One of the studies of practitioners was conducted in the UK in 2002–03 [43], and the other in the US (date of survey not reported) [2]. As shown in Table 1, studies used heterogeneous measures to classify CP, including ten describing CP as relating to "change" in gender identity or sexual orientation [2, 3, 8, 29, 35, 36, 39–41, 43], six

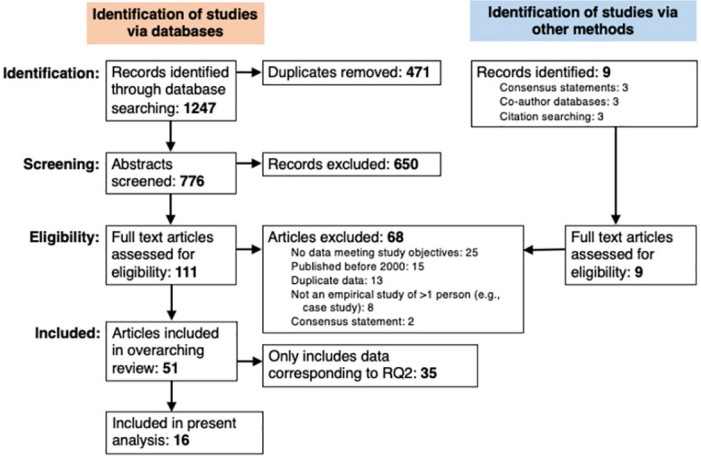

**Fig 1. Flowchart of studies screened and included in a systematic review of the prevalence of exposure to conversion practices among sexual and gender minority populations, internationally, 2000–2020.**

**Table 1. Characteristics of studies included in a systematic review of the prevalence of exposure to conversion practices among sexual and gender minority populations, internationally, 2000–2020.**

| First author (year of publication) | Geographic location | Year(s) of study | Sample population (study name) | Sampling methods | Definition of CP | Sample size* | Subgroup estimates |
|---|---|---|---|---|---|---|---|
| Bartlett (2009) [43] | United Kingdom | 2002–03 | Mental health practitioners belonging to one of 4 psychological, psychiatric, or counselling professional bodies | Random selection from professional registries | Ever having "assisted a client to reduce or change their same-sex desires" | 1406 | None |
| Blais (2022) [29] | Canada | 2019–20 | LGBTQI2+ people 18 + years of age (*Understanding the Inclusion and Exclusion of LGBTQ People*) | Social media, community partners, word-of-mouth | "In your lifetime, have you been involved, voluntarily or otherwise, in services to change your sexual orientation or to avoid being or becoming gay, lesbian, or bisexual? Or, to avoid becoming or being trans, to change your gender identity or gender expression, or to help you conform to the sex or gender assigned to you at birth?" | 3261 | Age; sexual orientation; gender modality; gender identity; race; education; income; immigration status; religiosity |
| Blosnich (2020) [3] | United States | 2016–17 | Cis sexual minorities, 18–59 years of age (*Generations Study*) | Random digit dial | "Did you ever receive treatment from someone who tried to change your sexual orientation (such as try to make you straight/heterosexual)?" | 1518 | Gender identity; sexual orientation; race/ethnicity; education |
| Del Rio-Gonzalez (2019) [40] | Colombia | 2019 | People 18+ years of age who identified as a sexual or gender minority | Social media (Facebook, Instagram), Pride events, snowball sampling | "Did you receive treatment from someone who tried to change your sexual orientation [such as to try to make you straight/heterosexual]?" And: "Have you ever received treatment from someone who tried to make you identify only with your sex assigned at birth [in other words, try to prevent you from being transgender]?" | 4160 | Gender identity; gender modality; sexual orientation |
| Government Equalities Office (2018) [42] | United Kingdom | 2017 | People 16+ years of age who identified as LGBT | Networks, Pride events, national media, social media | "Have you ever had so-called 'conversion' or 'reparative' therapy in an attempt to 'cure' you of being LGBT?" | 108100 | None |
| Green (2020) [17] | United States | 2018 | Young people 13–24 years of age who identified as LGBTQ | Advertising via Facebook and Instagram | "Have you ever undergone reparative therapy or conversion therapy?" | 25791 | Age; race/ethnicity; income; sexual orientation; gender modality |
| Higbee (2022) [36] | United States (Southern) | 2017–18 | The LGBTQ Institute Southern Survey | Advertising via social media and print | "During adolescence, were you ever sent to a therapist/ mental health practitioner, clergy/religious leader, or some other individual or organization in an effort to change your sexual orientation?" | 4096 | Age; race/ethnicity; gender modality; gender identity; sexual orientation; education; religiosity |

*(Continued)*

**Table 1.** (Continued)

| First author (year of publication) | Geographic location | Year(s) of study | Sample population (study name) | Sampling methods | Definition of CP | Sample size* | Subgroup estimates |
|---|---|---|---|---|---|---|---|
| Jones (2021) [39] | Australia | 2019 | LGBTQA+ young people, 14–21 years of age (*Writing Themselves in 4*) | Advertising via Facebook and Instagram | Experience "attending counselling, group work, programs or other interventions aimed at changing their sexuality or gender identity (formal conversion practices)" | 6418 | Gender identity; gender modality; religious affiliation; employment status |
| Lee (2021) [41] | South Korea | 2016 | Cis lesbian, gay, and bisexual people, 19+ years of age (*Rainbow Connection Project I—Korean Lesbian, Gay, & Bisexual Adults' Health Study*) | Advertising via Facebook and Twitter | "Have you ever received therapy or counseling to change your sexual orientation?" | 2168 | Gender identity; sexual orientation; age; urbanicity; education; income; employment |
| McGeorge (2015) [2] | United States | Not reported | Marriage and family therapists | Email invitation to members of professional registry | "Do you practice therapy intended to change sexual orientation from homosexual to heterosexual (i.e., reparative, conversion, or reorientation therapies)?" | 762 | None |
| Meanley (2019) [34] | United States (Baltimore, Chicago, Los Angeles, Pittsburgh) | 2016–17 | Cis men who have sex with men [60], 40+ years of age (*Multicenter AIDS Cohort Study*) | Media advertising, personal networks, gay medical clinics (cohort recruited in 1980s) | Any lifetime experience with any of the following 'conversion therapies': "psychotherapy, group-based therapy, prayer/religion-based therapy, gender role reinforcement, aversion therapy, pharmacological treatments, and other types" | 1238 | Race, HIV status, education |
| Ryan (2018) [35] | United States (California) | Not reported | White and Latino people 21–25 years of age who identified as LGBT and lived with a parent/guardian during adolescence | Local bars, clubs, community agencies | "Between ages 13 and 19, how often did any of your parents/caregivers take you to a therapist or religious leader to cure, treat, or change your sexual orientation?" | 245 | None |
| Salway (2020) [38] | Canada | 2011–12 | Sexual minority men, 15 + years of age (*SexNow Study*) | Sexual minority community venues (online) | "Have you ever attended sexual repair/reorientation counseling?" | 8388 | Sexual orientation; gender modality; age; race/ethnicity; urbanicity; province/territory; education; income |
| Salway (2021) [8] | Canada | 2019–20 | Sexual minority men, 15 + years of age (*SexNow Study*) | Sexual minority community venues (online) | CP defined as: "attempts to change sexual orientation or gender identity [including] more organized activities (such as counseling or faith-based rituals) that are sometimes referred to as 'conversion therapy'" | 9214 | Sexual orientation; gender modality; age; race/ethnicity; urbanicity; province/territory; education; income; immigration status |
| TransPulse (2020) [37] | Canada | 2019 | Trans people (identify with gender identity other than that assigned at birth), 14 + years of age | Community-based outreach | "Did you ever participate in any counselling or programs to try to make your gender match with your sex assigned at birth?" | 2033 | Age |

(*Continued*)

**Table 1.** (Continued)

| First author (year of publication) | Geographic location | Year(s) of study | Sample population (study name) | Sampling methods | Definition of CP | Sample size* | Subgroup estimates |
|---|---|---|---|---|---|---|---|
| Turban (2020) [18] | United States | 2015 | Trans adults (*2015 US Transgender Survey*) | Community-based outreach | "Did any professional (such as a psychologist, counselor, or religious advisor) try to make you identify only with your sex assigned at birth (in other words, try to stop you being trans)?" | 27715 | Gender identity; sexual orientation; race/ ethnicity; age; relationship status; education; employment; income |

Note.

* analytic sample size (i.e., only including those who responded to question regarding CP); cis = cisgender; CTP = conversion therapy practices; LGBTQ = lesbian, gay, bisexual, transgender, or queer; trans = transgender; gender modality refers to trans or cis

explicitly naming "conversion" [2, 8, 17, 34, 39, 42], four using the language of "repair" [2, 17, 38, 42], four defining CP as to make gender identity conform with gender/sex assigned at birth [18, 29, 37, 40], two using the language of "cure" [36, 42], two using the language of reorientation [2, 38], one using the language of reducing "same-sex attraction/desires" [43], and one using the language of "gender role enforcement" [34] (counts not mutually exclusive).

## Risk of bias

The count of items with high risk of bias ranged 0–4 (out of seven) across the 16 studies included (Table 2). Representativeness of a full national population varied, with eight of the 16 studies excluding sub-populations on the basis of age, geography, or gender, which we assessed as having high risk of bias. All but one study had high risk of non-response bias, owing to the lack of a defined source population for all of the non-probability-based SGM samples (and therefore an inability to estimate a response rate). Five of the 16 studies had high risk of bias with regard to the case definition; in these studies, no definition of CP was provided. Three studies used a CP recall period that was shorter than lifetime. Three studies combined different modalities of data collection (interviewer, online self-completed, or pen and paper self-completed). No studies had high risk of bias with regard to indirect data collection or inappropriate numerators/denominators.

## Prevalence estimates from studies of sexual and gender minority people

Prevalence estimates among SGM samples ranged from 2% [42] to 34% [35] (median: 8.5%; interquartile range: 4%, 13.5%). As anticipated, prevalence estimates were statistically heterogeneous ($I^2$ = 99.82%; $Q$[df = 12] = 6854.9779, $p$<0.0001). When stratified by covariate subgroups, median prevalence estimates differed by country, gender modality, gender modality combined with gender/sex assignment at birth, and race, but did not by sexual orientation (Table 3). The only countries with more than one available study were Canada and the US. The median prevalence estimate from studies in the US (N = 6; 13%) was approximately double the median estimate from studies in Canada (N = 4; 7%). Although there was only one study from Latin America, specifically Colombia, this study stands out as an outlier, estimating an overall prevalence of CP among SGM of 22% (Fig 2).

The median prevalence estimate from samples or subsamples of trans people (N = 10; 12%) was three times that from samples or subsamples of cis people (N = 11; 4%) (Table 3; S1 Fig). Stratified by both gender modality and gender/sex assignment at birth, median prevalence

**Table 2.** Risk of bias of studies included in a systematic review of the prevalence of exposure to conversion practices among sexual and gender minority populations, internationally, 2000–2020.

| First author (year of publication) | Representativeness (population segment) | Non-response bias | Direct data collection | Consistent modality | Acceptable case definition* | Appropriate prevalence period[†] | Appropriate numerators / denominators | Count of items with high risk of bias |
|---|---|---|---|---|---|---|---|---|
| Bartlett (2009) [43] | LOW | LOW | LOW | LOW | LOW | LOW | LOW | 0 |
| Blais (2022) [29] | HIGH (1 province) | HIGH | LOW | LOW | LOW | LOW | LOW | 2 |
| Blosnich (2020) [3, 61] | LOW | HIGH | LOW | LOW | LOW | LOW | LOW | 1 |
| Del Rio-Gonzalez (2019) [40] | LOW | HIGH | LOW | LOW | LOW | LOW | LOW | 1 |
| Government Equalities Office (2018) [42] | LOW | HIGH | LOW | LOW | HIGH | LOW | LOW | 2 |
| Green (2020) [17] | HIGH (<25 years of age) | HIGH | LOW | LOW | HIGH | LOW | LOW | 3 |
| Higbee (2022) [36] | HIGH (1 region) | HIGH | LOW | LOW | LOW | HIGH | LOW | 3 |
| Jones (2021) [39] | HIGH (<22 years of age) | HIGH | LOW | LOW | LOW | LOW | LOW | 1 |
| Lee (2021) [41] | LOW | HIGH | LOW | LOW | HIGH | LOW | LOW | 2 |
| McGeorge (2015) [2] | LOW | HIGH | LOW | LOW | LOW | HIGH | LOW | 2 |
| Meanley (2019) [34] | HIGH (40+ years of age; 4 cities) | HIGH | LOW | HIGH | HIGH | LOW | LOW | 4 |
| Ryan (2018) [35] | HIGH (21–25 years of age; 1 region) | HIGH | LOW | HIGH | LOW | HIGH | LOW | 4 |
| Salway (2020) [38] | HIGH (men) | HIGH | LOW | LOW | HIGH | LOW | LOW | 3 |
| Salway (2021) [8] | HIGH (men) | HIGH | LOW | LOW | LOW | LOW | LOW | 2 |
| TransPulse (2020) [37] | LOW | HIGH | LOW | HIGH | LOW | LOW | LOW | 2 |
| Turban (2020) [18, 62] | LOW | HIGH | LOW | LOW | LOW | LOW | LOW | 1 |

Note.

* Studies were assessed to have a high risk of bias if no definition of conversion practices was provided;

[†] studies were assessed to have a high risk of bias if they used any recall period other than lifetime experience.

estimates followed a gradient, with the lowest estimate among cis AFAB respondents (N = 7; 3%), greater estimates among trans and nonbinary AFAB respondents (N = 3; 5%) and among cis AMAB respondents (N = 8; 8%), and the greatest estimate among trans and nonbinary AMAB respondents (N = 3; 10%) (Table 3; Fig 3). Finally, the median prevalence estimates for Indigenous (N = 4; 8%) and racialized non-Indigenous (N = 8; 8%) people were nearly double that for white people (N = 8; 5%) (Table 3; S2 Fig).

Median prevalence estimates were comparable across all three sexual orientation sub-groups, i.e., asexual (N = 3; 4%), gay/lesbian (N = 9, 5%), and plurisexual (N = 9; 4%) (Table 3; S3 Fig). Several studies reported stratified CP prevalence estimates by age (Table 1); however, age categories were incongruous across studies and therefore could not be combined into summary measures. Among nine studies that examined age-related patterns across the full life course, two found no difference in CP prevalence by age at time of study [34, 38], three found greater CP prevalence among older SGM [3, 29, 42], two found greater CP prevalence among younger SGM people [8, 36], and two found a curvilinear relationship between age and CP prevalence, with the greatest prevalence observed among young adults, and lower prevalence observed among youth and older adults [18, 41].

**Table 3. Summary estimates (median, range, interquartile range) of prevalence estimates of exposure to conversion practices among sexual and gender minority populations, internationally, 2000–2020.**

| Covariate | Subgroup | Number of studies included | Median prevalence estimate | Range of prevalence estimates | IQR of prevalence estimates |
|---|---|---|---|---|---|
| Country | Canada | 4 | 0.07 | 0.04, 0.11 | 0.04, 0.10 |
|  | US | 6 | 0.13 | 0.04, 0.34 | 0.08, 0.17 |
| Gender modality | Cis | 11 | 0.04 | 0.02, 0.21 | 0.04, 0.10 |
|  | Trans | 10 | 0.12 | 0.04, 0.31 | 0.05, 0.16 |
| Gender modality and gender/sex assignment | Cis AMAB | 8 | 0.08 | 0.02, 0.20 | 0.05, 0.10 |
|  | Cis AFAB | 7 | 0.03 | 0.02, 0.22 | 0.02, 0.07 |
|  | Trans & nonbinary AFAB | 3 | 0.05 | 0.04, 0.28 | 0.05, 0.17 |
|  | Trans & nonbinary AMAB | 3 | 0.10 | 0.05, 0.32 | 0.08, 0.21 |
| Sexual orientation | Asexual | 3 | 0.04 | 0.03, 0.13 | 0.04, 0.09 |
|  | Gay/lesbian | 9 | 0.05 | 0.02, 0.23 | 0.04, 0.10 |
|  | Plurisexual | 9 | 0.04 | 0.02, 0.17 | 0.02, 0.07 |
| Race* | Indigenous to North America | 4 | 0.08 | 0.07, .014 | 0.07, 0.10 |
|  | Racialized non-Indigenous | 8 | 0.08 | 0.05, 0.29 | 0.06, 0.16 |
|  | White | 8 | 0.05 | 0.02, 0.13 | 0.04, 0.10 |

Note.

* only reported in studies from Canada, the UK, and the US. AFAB = assigned female at birth; AMAB = assigned male at birth; cis = cisgender; IQR = interquartile range; trans = transgender.

## Prevalence estimates from studies of practitioners

A 2002–03 study of N = 1328 British psychologists, psychiatrists, and counsellors found that 17% reported having "treated at least one client/patient in order to reduce or change his or her homosexual or lesbian feelings" [43]. This prevalence was highest among psychologists (21%)

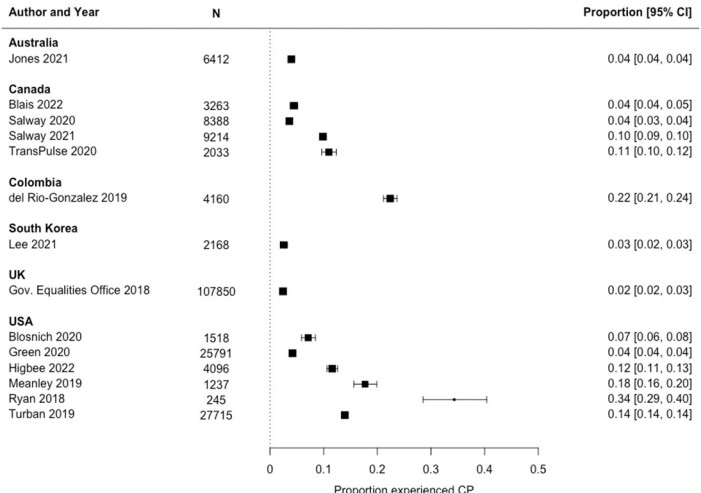

**Fig 2. Forest plot of lifetime prevalence of conversion practices (CP) among sexual and gender minority populations, stratified by country, 2000–2020.**

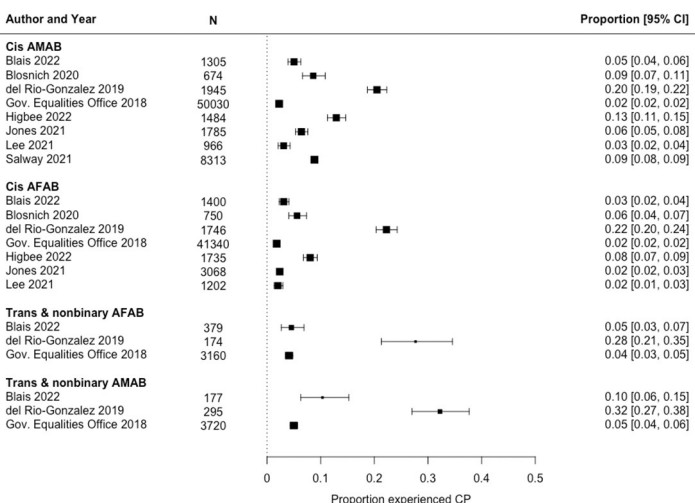

**Fig 3. Forest plot of lifetime prevalence of conversion practices (CP) among sexual and gender minority populations, stratified by gender modality and gender/sex assigned at birth, 2000–2020.**

and psychotherapists (18%), lower among counsellors (16%), and lowest among psychiatrists (11%) [43]. A second study, published in 2015, found that 4% of N = 762 marriage and family therapists who completed a survey in the US had practiced "conversion therapy" [2].

## Discussion

Synthesizing 16 empirical studies of SGM populations and mental health practitioners in six countries, over the past 20 years, we find that CPs remain prevalent, despite denouncements from healthcare professional bodies including the American Psychological Association, who first issued a resolution opposing CP in 1997 [24, 25]. In the current study, CP prevalence was found to range from 2% to 34% across 14 samples of SGM people and yielded a median estimate of 8.5%. CP prevalence estimated from two surveys of mental health practitioners (4%, 17%) fell within this range. We identified substantial heterogeneity in CP prevalence estimates, with some of this variability explained by theoretically important contextual and compositional covariates, including country, gender modality and gender/sex assignment at birth, and race.

With regard to country, prevalence estimates from the US (median 13%) were greater than those from Canada (median 7%). This difference may be interpreted as a function of comparatively more affirming social environments for SGM populations and stronger legal protections in Canada versus the US [44], though it is important to note that social attitudes [45]—and hence CP prevalence [8, 46]—vary meaningfully and are continuing to evolve within and across each of these countries. Differences in recruitment strategies, year of study, and framing of the questions may further explain cross-country differences in prevalence.

Analyses of CP prevalence by gender modality and gender/sex assignment at birth revealed two patterns. First, the prevalence of CP appears to be substantially greater among trans people (median 12%) than among cis people (median 4%). This finding is likely attributable to the longer history of health professionals denouncing CP targeting sexual orientation, as compared to the more recent denouncements of CP targeting gender identity [24], not to mention the slower improvements in affirming attitudes, social policies and legal protection regarding transgender people (as compared to cis gay and lesbian people) [24]. Second, among both trans and cis respondents, CP prevalence is greater among those assigned male at birth

(median 10% among trans and nonbinary AMAB; 8% among cis AMAB) than among those assigned female at birth (median 5% among trans and nonbinary AFAB; 3% among cis AFAB). While interpretation of this gradient requires nuance, it may at least partially reflect a broad and unjust societal preference for or privileging of masculinity over femininity (i.e., sexism) [47]. For instance, masculinity among AFAB children is often tolerated as a form of 'tomboyish' behaviours prior to puberty, whereas femininity among AMAB children is stigmatized as 'sissy' behaviour [48]. In other words, amid overlapping contexts of sexism, gender binarism, and cisheteropatriarchy [49], penalties for violating gender norms and expectations of masculinity may in this case be greater than those rendered for violating gender norms and expectations of femininity [50].

With respect to race, a higher prevalence was estimated for Indigenous (median 8%) and non-Indigenous racialized (median 8%) study respondents, as compared with white respondents (median 5%). In the case of Indigenous Peoples, this disparity may be the result of systemic, centuries-long campaigns on the part of European colonizers to force Indigenous Peoples—particularly children, through residential boarding schools—to conform with dominant European norms regarding gender and sexuality [51, 52]. Additional data are needed to interpret the disparity identified for non-Indigenous racialized people. In studies that provided CP estimates further disaggregated by specific racial categories, estimates tended to be highest for racialized participants identifying as African, Arab, Black, Caribbean, or Latin American/Latinx [3, 8, 17, 18].

Finally, with respect to age at time of study and sexual orientation, this systematic review could not discern consistent patterns in CP prevalence. Discrepant age-related patterns observed across the studies we included may be attributable to differences in age categorizations, time and place of study, and/or competing age and cohort-related forces [53]. Two conflicting hypotheses suggest complementary explanations for how CP prevalence may be expected to increase or decrease with age in a cross-sectional study. On the one hand, we may expect that older SGM would have experienced greater exposure to CP due to coming of age during a time when sexual and gender diversity were more pathologized by health professionals, as well as having had a greater cumulative number of years for the opportunity of exposure to CP [54]. On the other hand, there is no guarantee that broad improvements in social attitudes toward SGM will confer improvements in social environments. Hence, Russell & Fish have hypothesized a 'developmental collision' experienced by SGM who have increased periods of visibility during adolescence by virtue of coming out as trans or queer at a younger age than their older counterparts [55, 56]. In this context, we may even expect that contemporary youth have more 'opportunities' for exposure to CP. Finally, age-related differences may be attributable to differences in interpretation of or familiarity with the language of CP, whereby younger SGM may more readily identify their CP-related experiences as CP, while older SGM do not share the same lexicon and hence may deny having experienced CP, despite having endured these practices.

## Limitations

By limiting our searches to a finite set of bibliographic databases and websites, we acknowledge that we may have excluded perspectives, particularly those of historical nature. As noted in the opening of this article and demonstrated by the heterogeneity in language used to measure CP, there are numerous methodological challenges to synthesizing estimates of CP prevalence among SGM populations. First, we assumed that all studies were purporting to measure the same ill-defined set of practices (i.e., CP); however, it is likely that some of the variability in estimates between studies is attributable to information bias in how CP cases are captured.

Evidence of this can be found in two of the studies we included which are two iterations of the same survey/sample frame (i.e., the Canadian *SexNow* study) [8, 38]. In the 2011 iteration of *SexNow*, respondents were asked about experiences with "sexual repair/reorientation counseling"—a relatively narrow definition of CP that produced a relatively low prevalence estimate of 4% [38]. In the 2019–20 iteration of *SexNow*, respondents were asked about experiences with "attempts to change sexual orientation or gender identity [including] more organized activities (such as counseling or faith-based rituals) that are sometimes referred to as 'conversion therapy'"—a relatively broader definition yielding a far greater prevalence estimate of 10% [8]. This limitation is also reflected in the varying recall periods used, with three studies limiting the prevalence period to a particular age range (see Table 2).

Further limitations relate to sampling frames used in the studies included in this review and are reflected in the high risk of bias for non-representativeness and non-response across most of the studies included. We assumed a conceptual target population of all SGM people; however, this is operationalized differently across SGM samples, in some cases requiring an SGM identity (e.g., gay, lesbian, bisexual) or history of gender transition and in other cases including people who experience same-gender/sex attraction or a felt gender that differs from that assigned at birth [27, 28]. Individuals who experience CP and subsequently express cis and heterosexual identities are therefore unlikely to be included in the studies synthesized in this review. Ultimately, however, the heterogeneity of people who *could* be exposed to CP and lack of clear method for differentiating between those who are and are not potential targets of CP make it impossible to generate meaningful estimates that include cis and heterosexual people. Furthermore, we excluded studies with sampling frames and advertisements that intended to capture people actively seeking CP [22, 29]. If we had included these studies, we anticipate that the median prevalence estimates reported would be even higher (indeed, two of the studies excluded for this reason reported CP prevalence estimates of 15% [57] and 20% [22], both higher than the median of the 15 included studies).

Finally, with respect to data synthesis, we were unable to conduct meta-analysis due to the substantial heterogeneity in estimates, differing measures of CP, and compositional differences across samples. This in turn limited our ability to estimate the magnitude in differences between CP prevalence across subgroups [32].

## Implications

Based on the results of this systematic review, we formulate the following recommendations for researchers, policymakers, and other authorities. Future studies of CP prevalence would benefit from improved and consistently used definitions of CP. Where a local community-determined or jurisdiction-specific definition of CP is not evident, it may be useful for researchers to consider legal scholar Florence Ashley's definition for the sake of its balance between inclusivity (i.e., epidemiological sensitivity) of a broad set of CP and conversion-related practices and precision (i.e., epidemiological specificity) [24]. Their definition conceptualizes CP as "any treatment, practice, or sustained effort that aims to change, discourage, or repress a person's sexual orientation, gender identity, gender expression. . . or any behaviour associated with a gender other than the person's sex assigned at birth" [24]. We further call for more studies of CP focused on the SGM subgroups who carry the largest burden, including trans people, Indigenous Peoples, and non-Indigenous racialized groups (in 'western'/European settings, such as Canada, the US, and the UK). Given that CP occur in both religious and secular contexts [8, 29, 34], further research should investigate the degree to which particular religious ideologies continue to promote CP, as well as differences in the scope or nature of CP by religion.

Structural efforts to prevent CP need not wait for more evidence. Any amount of CP is worthy of policy interventions, and our finding that approximately 7% of SGM have been exposed to these practices should motivate concerted action on the part of governments and civil society groups. Given the much higher CP prevalence estimates among trans communities, it is critical for bans on CPs to cover gender identity and expression. Legislative bans are one important component to anti-CP strategies, though bans must be paired with other efforts and not rely solely on criminal enforcement. Such efforts may include enforced professional and regulatory standards for licensed practitioners who may be in a position to conduct CPs [58], promotion of SGM-affirming professional culture in healthcare and other settings where CPs occur (e.g., religious or faith-based settings) [24, 59], and broad, public education about the persistence of CPs and its harms [1]. Should this review be repeated in the future, we hope that prevalence will have abated; if not, these prevention strategies must be revisited.

## Supporting information

**S1 File. Sample search strategy (MEDLINE).**
(DOCX)

**S2 File. Dataset.**
(CSV)

**S1 Fig. Forest plot of lifetime prevalence of conversion practices (CP) among sexual and gender minority populations, stratified by gender modality, 2000–2020.**
(TIF)

**S2 Fig. Forest plot of lifetime prevalence of conversion practices among sexual and gender minority populations, stratified by race, 2000–2020.**
(TIF)

**S3 Fig. Forest plot of lifetime prevalence of conversion practices (CP) among sexual and gender minority populations, stratified by sexual orientation, 2000–2020.**
(TIF)

**S1 Checklist. PRISMA 2020 main checklist.**
(DOCX)

## Acknowledgments

We are grateful to the survivors of conversion practices who have courageously shared their stories to inspire action on the part of researchers, policymakers, and others.

## Author Contributions

**Conceptualization:** Travis Salway, David J. Kinitz, Hannah Kia, Florence Ashley, Dean Giustini, Olivier Ferlatte, Trevor Goodyear, Alex Abramovich.

**Data curation:** Travis Salway, Florence Ashley, Dean Giustini, Amrit Tiwana, Reilla Archibald, Amirali Mallakzadeh, Elisabeth Dromer.

**Formal analysis:** Travis Salway, Dean Giustini, Amrit Tiwana.

**Funding acquisition:** Travis Salway.

**Investigation:** Travis Salway, Hannah Kia, Florence Ashley, Dean Giustini, Reilla Archibald, Amirali Mallakzadeh, Elisabeth Dromer, Trevor Goodyear.

**Methodology:** Travis Salway, David J. Kinitz, Hannah Kia, Florence Ashley, Olivier Ferlatte, Alex Abramovich.

**Project administration:** Travis Salway.

**Resources:** Travis Salway, Dean Giustini, Olivier Ferlatte.

**Supervision:** Travis Salway.

**Validation:** Dean Giustini.

**Visualization:** Travis Salway.

**Writing – original draft:** Travis Salway, David J. Kinitz, Hannah Kia, Florence Ashley, Dean Giustini, Amrit Tiwana, Reilla Archibald, Trevor Goodyear.

**Writing – review & editing:** Travis Salway, David J. Kinitz, Hannah Kia, Florence Ashley, Dean Giustini, Amrit Tiwana, Reilla Archibald, Amirali Mallakzadeh, Elisabeth Dromer, Olivier Ferlatte, Trevor Goodyear, Alex Abramovich.

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
