## [Decision Letter · Decision Letter 0]

6 Jun 2023

PONE-D-23-01755A systematic review of the prevalence of lifetime experience with ‘conversion’ practices among sexual and gender minority populationsPLOS ONE

Dear Dr. Salway,

Thank you for submitting your manuscript to PLOS ONE. After careful consideration, we feel that it has merit but does not fully meet PLOS ONE’s publication criteria as it currently stands. Therefore, we invite you to submit a revised version of the manuscript that addresses the points raised during the review process.

We look forward to receiving your revised manuscript.

Kind regards,

Angélica Baptista Silva, Ph.Sc.

Academic Editor

PLOS ONE

Journal Requirements:

"This study was funded by the Canadian Institutes of Health Research, which had no role in the design of the review, data collection or analysis, or decision to publish."

"TS, Canadian Institutes of Health Research (PCS - 168193)

https://cihr-irsc.gc.ca/

Reviewers' comments:

Reviewer's Responses to Questions

**Comments to the Author**

1. Is the manuscript technically sound, and do the data support the conclusions?

Reviewer #1: Yes

Reviewer #2: Yes

2. Has the statistical analysis been performed appropriately and rigorously? 

Reviewer #1: N/A

Reviewer #2: Yes

3. Have the authors made all data underlying the findings in their manuscript fully available?

Reviewer #1: Yes

Reviewer #2: Yes

4. Is the manuscript presented in an intelligible fashion and written in standard English?

Reviewer #1: Yes

Reviewer #2: Yes

5. Review Comments to the Author

Reviewer #1: really interesting job. I have suggestions to make.

1) In introduction part before to focus on CP practice please explore briefly the main problems that transgender people could copy in our society. For example in health care (Lavorgna L, Moccia M, Russo A, Palladino R, Riccio L, Lanzillo R, Brescia Morra V, Tedeschi G, Bonavita S. Health-care disparities stemming from sexual orientation of Italian patients with Multiple Sclerosis: A cross-sectional web-based study. Mult Scler Relat Disord. 2017 Apr;13:28-32. doi: 10.1016/j.msard.2017.02.001. Epub 2017 Feb 5. PMID: 28427697.) or from economic point of view.

2) please revise the English of the manuscript

3) please add a graphical summary

Reviewer #2: Paper review

The paper “A systematic review of the prevalence of lifetime experience with ‘conversion’ practices among sexual and gender minority populations” it's a challenging writing with a complex subject of study, that is focused on SGM people who had experienced CP in different countries around the world. I think it has a major issue that deserves all the support and visibility from the scientific community, mainly from the psychiatric and psychological researchers that deal with mental health within SGM´s groups.

I wrote a review separately for each section of the text and focused on bringing the positive aspects of the research and pointing out some possibilities of improvement.

Introduction

Here language is clear and attends to technical needs with no many difficult words. The introduction is very detailed and gives to the reader a complete description on the methods and shows the unique aspects and relevance of this paper, it does a really good job of giving a general idea to the whole text.

Something that could make the beginning a little more interesting for those who are reading would be to mention some data about gender and sexual diversity among the writers of the article. SGM people that are researchers and are reading the paper can create more boundaries to the research if they get to see diversity.

Methods

The description of the methods is well structured. I really liked that it mentioned the use of references contemplating SGM writers, because this increases visibility and helps break up with stigma.

Even though there are not relevant studies to the research that provides data about the economical aspects of SGM who experienced CP, this aspect could be a little more contextualized to the economical aspects of the countries in general. I´m pointing this out because a part of the SGM population, such as transgender people, a part of this community has needs on body modifications, to affirm gender and decrease dysphoria feelings some may proceed with surgeries, to put implants, to change the voice and other aspects... However, to access these procedures with safe medical circumstances guaranteed it's very expensive, at the same time, most of the countries do not offer the procedures in their public health systems.

“Risk of bias”, is a concept that I never had heard of before reading this paper. This concept could be more detailed and explained, maybe it would be interesting to mention why the group chose Hoy´s method for this research.

Analysis

This is the most complex part of the article. I understand the need of using all the scientific terms and concepts. However the writing here contains too many unusual words with a very specific vocabulary, this kind of language can become a challenge for researchers and readers that are not native english speakers.

Results

The way the authors present the results of this study is very satisfactory because it illustrates in many aspects the experiences that SGM faces during CP incidences. Sadly, the data presented reveals how oppressive and violent the manifestations of CP can be. One very positive aspect is the critical observation on how some of the articles do not define CP and how this can increase risk of bias and stigma between SGM groups.

Discussion

The discussion on the topic is one of the most important parts of this paper, because it is the part where the authors compare some features of the research based on the findings of the data analysis.

The limitations were well described and justified with the complexity of the subject.

Implications

I suggest that in this topic, a more detailed analysis on the CP´s link to the religious groups or practices, and by doing so an intersectional relation with transgender community and gay man. It has been known that many religious centers can be responsible for creating violent processes to gender non-conforming groups and sexual minorities. Would be interesting to investigate the basis of the religions that are mentioned in these studies and point out who they are. If they are Catholic, Protestants, Jewish, or any other…

6. PLOS authors have the option to publish the peer review history of their article (what does this mean?). If published, this will include your full peer review and any attached files.

Reviewer #1: No

Reviewer #2: **Yes: **Raphaellie Lázaro Rezende Silva Maciel

---

## [Author Response · Author response to Decision Letter 0]

20 Jul 2023

Dear Dr. Baptista Silva,

We thank you and the reviewers for your thoughtful feedback on our manuscript, “A systematic review of the prevalence of lifetime experience with ‘conversion’ practices among sexual and gender minority populations.” We have edited the manuscript in response to these comments, as detailed below.

Sincerely,

Travis Salway & coauthors

Editor comments:

Comment E-1: Upon re-submitting your revised manuscript, please upload your study’s minimal underlying data set as either Supporting Information files or to a stable, public repository and include the relevant URLs, DOIs, or accession numbers within your revised cover letter.

Response: We have uploaded the dataset as a .CSV file.

Edit in manuscript file: “The data used in this systematic review are available in a Supplementary File (“Dataset”).” (p28)

Comment E-2: We note that you have provided funding information that is not currently declared in your Funding Statement. However, funding information should not appear in the Acknowledgments section or other areas of your manuscript. We will only publish funding information present in the Funding Statement section of the online submission form. 

Response: We have removed funding information from the Acknowledgments section of the manuscript.

Please update our Funding Statement as follows:

“This study was funded by the Canadian Institutes of Health Research (PCS – 168193). The funders had no role in study design, data collection and analysis, decision to publish, or preparation of the manuscript.”

Reviewer comments:

Comment R1-1: In introduction part before to focus on CP practice please explore briefly the main problems that transgender people could copy in our society. For example in health care (Lavorgna L, Moccia M, Russo A, Palladino R, Riccio L, Lanzillo R, Brescia Morra V, Tedeschi G, Bonavita S. Health-care disparities stemming from sexual orientation of Italian patients with Multiple Sclerosis: A cross-sectional web-based study. Mult Scler Relat Disord. 2017 Apr;13:28-32. doi: 10.1016/j.msard.2017.02.001. Epub 2017 Feb 5. PMID: 28427697.) or from economic point of view.

Response: We agree that we should contextualize CP with broader systemic factors (like cissexism) before describing the methodological issues in studying it. We have added a paragraph to the beginning of the manuscript, as recommended.

Edit in manuscript file: “CPs occur within a broader societal context of cissexism and heterosexism—belief systems that position cisgender and heterosexual identities and orientations as preferred and desirable.1 Cissexism and heterosexism lead to stigma, discrimination, violence, and a lack of protections and access to social determinants of good health for lesbian, gay, bisexual, transgender (trans), and queer (herein, sexual and gender minority [SGM]) people.1 This, in turn, produces numerous health disparities among SGM people, including unequal access to healthcare—a resource to which access in many settings is already limited by cost.4” (p4)

Comment R1-2: please revise the English of the manuscript

Response: Following reviewer 2’s comments regarding the Methods (specifically Analysis) section of the manuscript, we have added some clarifying language, which we hope improves the accessibility of the manuscript.

We are happy to make additional edits if the reviewer or editor have specific recommendations or requests.

Edit in manuscript file: See edits and parenthetical additions to Methods. (pp10-11)

Comment R1-3: please add a graphical summary

Response: We were unable to find instructions for graphical abstracts in the PLoS One submission guidelines so will defer to the editor.

Comment R2-1: Here language is clear and attends to technical needs with no many difficult words. The introduction is very detailed and gives to the reader a complete description on the methods and shows the unique aspects and relevance of this paper, it does a really good job of giving a general idea to the whole text.

Something that could make the beginning a little more interesting for those who are reading would be to mention some data about gender and sexual diversity among the writers of the article. SGM people that are researchers and are reading the paper can create more boundaries to the research if they get to see diversity.

Response: Thank you for the encouraging feedback regarding the Introduction.

We have added a brief author positionality statement to the first paragraph of the Methods section.

Edit in manuscript file: “We came to this study as an interdisciplinary group of health researchers invested in advancing equity- and evidence-informed public policy with SGM people, including in the context of CPs. Our authorship team includes SGM people and allies with expertise in population approaches to SGM health research.1” (p6)

Comment R2-2: The description of the methods is well structured. I really liked that it mentioned the use of references contemplating SGM writers, because this increases visibility and helps break up with stigma.

Response: Thank you for the feedback.

Comment R2-3: Even though there are not relevant studies to the research that provides data about the economical aspects of SGM who experienced CP, this aspect could be a little more contextualized to the economical aspects of the countries in general. I´m pointing this out because a part of the SGM population, such as transgender people, a part of this community has needs on body modifications, to affirm gender and decrease dysphoria feelings some may proceed with surgeries, to put implants, to change the voice and other aspects... However, to access these procedures with safe medical circumstances guaranteed it's very expensive, at the same time, most of the countries do not offer the procedures in their public health systems. 

Response: We have added a paragraph to the start of the manuscript to contextualize CP, and we have included within this paragraph an acknowledgment of the cost-related barriers to gender-affirming care. 

Edit in manuscript file: “CPs occur within a broader societal context of cissexism and heterosexism—belief systems that position cisgender and heterosexual identities and orientations as preferred and desirable.1 Cissexism and heterosexism lead to stigma, discrimination, violence, and a lack of protections and access to social determinants of good health for lesbian, gay, bisexual, transgender (trans), and queer (herein, sexual and gender minority [SGM]) people.1 This, in turn, produces numerous health disparities among SGM people, including unequal access to healthcare—a resource to which access in many settings is already limited by cost.4” (p4)

Comment R2-4: “Risk of bias”, is a concept that I never had heard of before reading this paper. This concept could be more detailed and explained, maybe it would be interesting to mention why the group chose Hoy´s method for this research. 

Response: We have added a definition of risk of bias, along with an explanation for our choice of the Hoy tool. There are few risk of bias tools available specifically for prevalence studies, and this one was the most flexible of those available. 

Edit in manuscript file: “We assessed risk of bias among the studies included in the review in order to systematically describe methodological limitations to the available literature. We used the Hoy et al. risk of bias tool for prevalence studies, given its flexibility across research contexts.26” (p10)

Comment R2-5: This is the most complex part of the article. I understand the need of using all the scientific terms and concepts. However the writing here contains too many unusual words with a very specific vocabulary, this kind of language can become a challenge for researchers and readers that are not native english speakers. 

Response: Thank you for this feedback. We have added definitions and clarifying explanations to this section. See edits and parenthetical additions to Methods. (pp10-11)

Comment R2-6: The way the authors present the results of this study is very satisfactory because it illustrates in many aspects the experiences that SGM faces during CP incidences. Sadly, the data presented reveals how oppressive and violent the manifestations of CP can be. One very positive aspect is the critical observation on how some of the articles do not define CP and how this can increase risk of bias and stigma between SGM groups. 

Response: Thank you for this feedback.

Comment R2-7: The discussion on the topic is one of the most important parts of this paper, because it is the part where the authors compare some features of the research based on the findings of the data analysis.

The limitations were well described and justified with the complexity of the subject.

Response: Thank you for this feedback.

Comment R2-8: I suggest that in this topic, a more detailed analysis on the CP´s link to the religious groups or practices, and by doing so an intersectional relation with transgender community and gay man. It has been known that many religious centers can be responsible for creating violent processes to gender non-conforming groups and sexual minorities. Would be interesting to investigate the basis of the religions that are mentioned in these studies and point out who they are. If they are Catholic, Protestants, Jewish, or any other…

Response: We have added a recommendation to this effect. 

Edit in manuscript file: “Given that CP occur in both religious and secular contexts8,29,35, further research should investigate the degree to which particular religious ideologies continue to promote CP, as well as differences in the scope or nature of CP by religion.” (p26)

---

## [Decision Letter · Decision Letter 1]

5 Sep 2023

A systematic review of the prevalence of lifetime experience with ‘conversion’ practices among sexual and gender minority populations

PONE-D-23-01755R1

Dear Dr. Salway,

We’re pleased to inform you that your manuscript has been judged scientifically suitable for publication and will be formally accepted for publication once it meets all outstanding technical requirements.

Kind regards,

Emily Lund

Academic Editor

PLOS ONE

Additional Editor Comments (optional):

Reviewers' comments:

Reviewer's Responses to Questions

**Comments to the Author**

1. If the authors have adequately addressed your comments raised in a previous round of review and you feel that this manuscript is now acceptable for publication, you may indicate that here to bypass the “Comments to the Author” section, enter your conflict of interest statement in the “Confidential to Editor” section, and submit your "Accept" recommendation.

Reviewer #2: All comments have been addressed

2. Is the manuscript technically sound, and do the data support the conclusions?

Reviewer #2: Yes

3. Has the statistical analysis been performed appropriately and rigorously? 

Reviewer #2: I Don't Know

4. Have the authors made all data underlying the findings in their manuscript fully available?

Reviewer #2: Yes

5. Is the manuscript presented in an intelligible fashion and written in standard English?

Reviewer #2: Yes

6. Review Comments to the Author

Reviewer #2: I have no more suggestions to make, since the author had made changes and added new information that was required.

7. PLOS authors have the option to publish the peer review history of their article (what does this mean?). If published, this will include your full peer review and any attached files.

Reviewer #2: **Yes: **Raphaellie Lázaro Rezende Silva Maciel

---

## [Editor Report · Acceptance letter]

12 Sep 2023

PONE-D-23-01755R1 

A systematic review of the prevalence of lifetime experience with ‘conversion’ practices among sexual and gender minority populations 

Dear Dr. Salway:

I'm pleased to inform you that your manuscript has been deemed suitable for publication in PLOS ONE. Congratulations! Your manuscript is now with our production department. 

Kind regards, 

on behalf of

Dr. Emily Lund 

Academic Editor

PLOS ONE